# *Verticillium dahliae VdPBP1* Transcription Factor Is Required for Hyphal Growth, Virulence, and Microsclerotia Formation

**DOI:** 10.3390/microorganisms12020265

**Published:** 2024-01-26

**Authors:** Huong Thi Nguyen, Thanh Thi Duong, Vu Xuan Nguyen, Tien-Dung Nguyen, Thuc Tri Bui, Dung Thuy Nguyen Pham

**Affiliations:** 1Faculty of Biotechnology, Thai Nguyen University of Sciences, Thai Nguyen 24000, Vietnam; huongnguyenthi@tnus.edu.vn; 2Faculty of Biotechnology and Food Technology, Thai Nguyen University of Agriculture and Forestry, Thai Nguyen 24000, Vietnam; duongthithanhk50cnsh@gmail.com (T.T.D.); nguyenxuanvu@tuaf.edu.vn (V.X.N.); dungnt@tuaf.edu.vn (T.-D.N.); 3NTT Institute of Applied Technology and Sustainable Development, Nguyen Tat Thanh University, Ho Chi Minh City 70000, Vietnam; 4Faculty of Environmental and Food Engineering, Nguyen Tat Thanh University, Ho Chi Minh City 70000, Vietnam

**Keywords:** *Verticillium* wilt disease, *VdPBP1*, fungal growth, microsclerotia, pathogenicity

## Abstract

*Verticillium dahliae*, a fungal pathogen that affects more than 200 plant species, including tomatoes, requires specific proteins for its early steps in plant infection. One such crucial protein, *VdPBP1*, exhibits high expression in the presence of tomato roots. Its 313-amino acid C-terminal section restores adhesion in nonadhesive *Saccharomyces cerevisiae* strains. To uncover its role, we employed a combination of bioinformatics, genetics, and morphological analyses. Our findings underscore the importance of *VdPBP1* in fungal growth and pathogenesis. Bioinformatic analysis revealed that the *VdPBP1* gene consists of four exons and three introns, encoding a 952-codon reading frame. The protein features a 9aaTAD domain, LsmAD, and PAB1 DNA-binding sites, as well as potential nuclear localization and transmembrane helix signals. Notably, the deletion of a 1.1 kb fragment at the gene’s third end impedes microsclerotia formation and reduces pathogenicity. Mutants exhibit reduced growth and slower aerial mycelial development compared to the wild type. The *VdPBP1* deletion strain does not induce disease symptoms in tomato plants. Furthermore, *VdPBP1* deletion correlates with downregulated microsclerotia formation-related genes, and promoter analysis reveals regulatory elements, including sites for Rfx1, Mig1, and Ste12 proteins. Understanding the regulation and target genes of *VdPBP1* holds promise for managing *Verticillium* wilt disease and related fungal pathogens.

## 1. Introduction

*Verticillium dahliae*, a fungal pathogen, inflicts root diseases on various plant species, causing wilt and early senescence in over 200 plants, including vital crops like tomatoes, potatoes, and strawberries [1,2,3]. This highly adaptable fungus is prevalent worldwide, including in Vietnam, and can survive in the soil for up to 15 years in the absence of a host, making its control challenging [1,2]. Crop rotation does not effectively mitigate the impact of this fungal disease on crops. Consequently, finding new methods to combat fungal diseases, specifically this pathogenic fungus, has become a significant concern globally [1,2]. However, to develop effective inhibitory measures, a comprehensive understanding of this fungal disease’s pathogenic and growth mechanisms is crucial. In *Saccharomyces cerevisiae* (*S. cerevisiae*), PAB1-binding protein 1 is involved in pre-mRNA polyadenylation. It may act to repress the ability of PAB1 to negatively regulate polyadenylation and function as a negative regulator of poly(A) nuclease (PAN) activity [4,5]. It also plays a role in promoting mating-type switching in mother cells by positively regulating HO mRNA translation. Disruption of this gene leads to abnormal mRNA processing, as observed in the disruption phenotype. In *Cryptococcus neoformans*, PAB1-binding protein 1 is involved in the post-transcriptional regulation of gene expression through interactions with poly A-binding protein. Disruption of the PAB1 gene results in abnormal sexual reproduction, and the mutant strain exhibits decreased virulence [6,7]. A counterpart to PAB1 in *V. dahliae*, known as the VdPBP1 protein, has been identified. Comparative analysis of *VdPBP1* gene expression in both standard MS medium and the MS environment within tomato plants revealed a significant upregulation in its expression. Additionally, it is known that the ORF harboring 313 amino acids at the protein’s C-terminal can restore adhesion in the *FLO8* mutant [3], a transcription factor responsible for regulating the expression of flocculation genes in *S. cerevisiae* [8,9]. This indicates a potential role for the *VdPBP1* gene in early host infection stages, such as root attachment and resistance to root toxins. Furthermore, the ORF at the C-terminal end of this protein may have a vital function. Nonetheless, the role of this gene in pathogenic fungi, particularly *Verticillium* fungi, remains unstudied.

## 2. Materials and Methods

### 2.1. Bioinformatic Analysis of DNA and Protein Sequences

Predicted conserved domains/motifs were examined with InterProScan 5 [10], Pfam [11], and Blast searches at the National Center for Biotechnology Information (NCBI blast) (http://www.ncbi.nlm.nih.gov (accessed on 20 June 2023)) [12] using the nine-amino-acid transactivation domain prediction [13]. Nuclear localization signals (NLS) were predicted using SeqNLS: nuclear localization signal prediction based on frequent pattern mining and linear motif scoring (http://mleg.cse.sc.edu/seqNLS/ (accessed on 20 June 2023)) [14]. Gene numbers and gene sequences of *V. dahliae* JR2 were identified according to the Ensemble Genomes [15] (http://fungi.ensembl.org/Verticillium_dahliaejr2/Info/Index (accessed on 20 June 2023)). Protein alignments were performed by Crustal Omega at the European Molecular Biology Laboratory’s European Bioinformatics Institute (http://www.ebi.ac.uk/Tools/msa/clustalo/ (accessed on 20 June 2023)) [16]. Prediction of the transmembrane helix was achieved through the utilization of the MEMSAT-SVM method developed by Nugen and Jones [17].

### 2.2. Microorganism Cultivation Condition

*Escherichia coli* strain DH5α (Invitrogen, Karlsruhe, Germany) and *Agrobacterium tumefaciens* strain AGL-1 [18] were used for transformation procedures in this study. They were cultivated in Luria–Bertani (LB) medium (0.5% yeast extract, 1% bacto-tryptone, and 1% NaCl) at 37 °C for *E. coli* or 25 °C for *A. tumefaciens*.

*V. dahliae* strain JR2 was provided by Bart Thomma, Laboratory of Phytopathology in Wageningen, the Netherlands (Table 1) [19]. All strains were inoculated at 25 °C in potato dextrose broth (PDA) (Sigma-Aldrich Chemie GmbH, Munich, Germany), minimal medium (MM), Czapek-Dox medium (CDM) [20], or a modified simulated xylem medium (SXM) [21] composed of 0.2% pectin from citrus peel (Sigma-Aldrich Chemie GmbH, Munich, Germany), 0.4% casein hydrolysate (OXOID Ltd., Basingstoke, Hampshire, UK), 2 mM MgSO_4_, 1× AspA (3.5 M NaNO_3_, 350 mM KCl, 560 mM KH_2_PO_4_, pH 5.5 with KOH), and 1× trace elements (5 g L^1^ FeSO_4_ × 7H_2_O, 50 g L^−1^ EDTA, 22 g L^−1^ ZnSO_4_ × 7H_2_O, 11 g L^−1^ H_3_BO_3_, 5 g L^−1^ MnCl_2_ × 4H_2_O, 1.6 g L^−1^ CoCl_2_ × 6H_2_O, 1.6 g L^−1^ CuSO_4_ × 5H_2_O, 1.1 g L^−1^ (NH_4_)_6_Mo_7_O_24_ × 4H_2_O; pH 6.5 with KOH).

The *Verticillium* strains were grown in liquid SXM on a shaker at 120 rpm at 25 °C for 7 days. Conidia were harvested by filtration of the culture through a Miracloth membrane (Merck KGaA, Darmstadt, Germany); the filtrate was washed twice with sterile water before resuspending in 0.96% NaCl and 0.05% tween 80. The number of spores was counted in a counting chamber under a binocular microscope, and the spore density was adjusted to 107 spores mL^−1^. Aliquots of spore suspension containing 25% glycerol were frozen in liquid nitrogen and stored at −80 °C.

### 2.3. Localization Studies

A fragment containing a putative ORF (open reading frame) of *VdPBP1#1*, which spans 939 bp without a stop codon, was amplified through a PCR reaction using the primer pair *OE-VdPBP1-F*/*OE-VdPBP1-R* (Table 2 and Table 3). The fragment was subsequently connected to the N-terminal region of a *GFP* gene in the pGreen2 vector [3] using a linker, resulting in the creation of pGreen2-*VdPBP1*#1 (Table 2). The pGreen2-*VdPBP1*#1 construct was introduced into the JR2-wt strain through *Agrobacterium tumefaciens*-mediated transformation, as described by Timpner et al. in 2013 [22]. Transformants were chosen on PDA plates supplemented with hygromycin (50 μg mL^−1^) and cefotaxime (100 μg mL^−1^). Screening for green fluorescence (GFP) was conducted under a fluorescent microscope. Localization was assessed using fluorescence microscopy, with fungal vesicles visualized through staining with FM4-64.

### 2.4. Gene Deletion

The vector pKO2 [23] was digested with enzymes *EcoR*I and *Hind*III. In parallel, the nourseothricin resistance cassette (1.45 kb) was amplified by PCR from the pNAT1 vector [24] with the primer pair gdpA-NAT-F/gdpA-NAT-R (Table 3). Fragments of 1.5 kb upstream and downstream of a putative ORF spanning 1.1 kb, known as *VdPBP1*#1 at the C-terminal end of the *VdPBP1* gene were amplified from the genomic DNA of the *V. dahliae* JR2 wild type (JR2-wt) using Phusion high-fidelity DNA polymerase (Thermo Fisher Scientific GmbH, St. Leon-Rot, Germany) with primer pairs *VdPBP1*-P1/*VdPBP1*-P2 and *VdPBP1*-P3/*VdPBP1*-P4. These primers harbored a 17-nucleotide overhang with a complementary sequence that matched the upstream and downstream on the recombination plasmid map (Table 3). 

The ligation of the DNA segment into the vector will be accomplished through the self-cutting and joining reaction facilitated by T4 DNA polymerase. The ligation reaction using T4 DNA polymerase includes the following components: 200 ng of the 6300 bp DNA plasmid segment, which has been previously cut with *EcoR*I and *Hind*III restriction enzymes; 200 ng of the 1.45 kb nourseothricin resistance cassette; DNA segments *VdPBP1*-F1 and *VdPBP1*-F2 (100 ng each); 1 µL of BSA; 1 µL of NEB 2 buffer; and 0.2 µL of T4 DNA polymerase. The thermal cycling for the ligation reaction is as follows: initial denaturation at 94 °C for 1 min, followed by annealing at 50 °C for 30 s, and, finally, cooling on ice for 10 min before being transferred into *E. coli*-competent cells using the heat shock method [25].

The recombination plasmid pKO2-*VdPBP1* was confirmed by PCR and digestion with restriction enzymes before introducing into the JR2-wt strain via *Agrobacterium tumefaciens* AGL-1 [18,22] to generate Δ *VdPBP1* (Table 1). The fungal transformants were selected on potato dextrose agar (PDA) containing nourseothricin (50 µg mL^−1^) and cefotaxime (100 µg mL^−1^). The deletion strains were confirmed by Southern hybridization.

### 2.5. Tomato Infection Assay

Tomato infection assay was modified from [24]. Ten-day-old tomato seedlings “MoneyMaker” (Bruno, Nebelung Gmb, Everswinkel, Germany) were infected with 10 mL of 2 × 10^6^ fungal spores per ml of the JR2-wt, deletion strains, and complementation strains or noninfected with tap water for 30 min by root dipping. Then, they were transferred into pots containing a mixture of sand and soil (1:1). Tomato plants were kept in the dark for 24 h to reduce stress effects. The infected plants were cultivated in a climate chamber with 16 h light and 8 h dark at 22–25 °C. The soil humidity was controlled at a minimum level to increase pathogenic symptoms. The percentage of disease index was calculated as described by Zhu et al. in 2013 [26]. The plant heights were measured weekly until 21 dpi. The fungal re-isolation from infected plants was modified from [24]. Hypocotyls of infected plants were sterilized in 6% hypochlorite and 0.05% tween solution for 7 min and 70% ethanol for 5 min, washed twice with distilled water for 5 min, and put on PDA plates containing chloramphenicol (100 μg mL^−1^) and cefotaxime (100 μg mL^−1^) to re-isolate the fungus. The plates were cultivated at 25 °C for 7 days. The total DNA was extracted from roots and stems. The experiments were repeated twice with 12 plants per treatment. 

### 2.6. Microsclerotia Counting

Equal quantities of spores from the JR2-wt, deletion, and complementation strains of *VdPBP1* were introduced onto a 3% cellulose (*w*/*v*) medium. The cultures were then incubated at 25 °C for a duration of seven days. Microsclerotia counts on the agar surface and within the agar were determined by employing binocular microscopy (Leica M165 FC, LEICA, Wetzlar, Germany) across three separate replicates at 5 days post-inoculation (dpi) and 7 dpi.

## 3. Results

### 3.1. The Gene VdPBP1 Encodes Proteins Comprising Both a LsmAD Domain and a PAB-Binding Domain

To identify homologs in other fungi and predict the domains and nuclear localization signal (NLS), the *VdPBP1* sequences were analyzed using NCBI blast, Pfam, 9aaTAD, and SeqNLS tools (Figure 1).

By aligning the cDNA and gDNA sequences of *VdPBP1*, it was observed that the gene structure consists of four exons and three introns, resulting in an open reading frame of 952 codons for a protein (refer to Figure 1).

The bioinformatic analysis results revealed that the *VdPBP1* sequence contains a 9aaTAD domain located in the middle of the protein sequence. Additionally, the N-terminus of the protein harbors two DNA-binding sites, namely LsmAD and PAB1 domains. Moreover, a potential nuclear localization signal (NLS) and a transmembrane helix are predicted using the method (MEMSAT-SVM) of Nugent and Jones [11]. Orthologs to the LsmAD domain were found in *Colletotrichum graminicola* (XP_008096966), *Cavenderia fasciculata* (EGG21785), *Heterostelium album* (EFA76666), *Dictyostelium purpureum* (EGC33099), *Dictyostelium discoideum* (Q55DE7), *Paramecium tetraurelia* (A0BD44), *Capsaspora owczarzaki* (XP_004342993), *Tetrahymena thermophila* (Q23QL5), *Leishmania infantum* (XP_001463764), *Trypanosoma cruzi* (Q4D0Q9), *Plasmodium yoelii* (Q7RRV2), and *Cryptosporidium muris* (XP_002139652).

### 3.2. VdPBP1 Is a Transmembrane Helix Protein

The VdPbp1 sequences contain a potential nuclear localization signal (NLS) PKGEAPK at the N-terminal and a transmembrane helix signal SPAQVPYSQPMMQPYG at the C-terminal. To investigate the localization of the VdPbp1 protein, a localization experiment was conducted. After 24 h of growth in SXM, the GFP signal from the transformed strains was observed using fluorescence microscopy. The fungal cell membrane was stained with FM4-64 (Figure 2b).

In strains carrying *VdPBP1#1*::GFP (Figure 2b), a strong GFP signal was observed in the cell membrane, accompanied by some large round dots. This GFP signal was colocalized with the vesicles stained by FM4-64 (Figure 2b). In contrast, the GFP signal was distributed throughout the entire hyphae in control cells expressing only free *GFP* (JR2-GFP), and it was not detected in the wild-type strain (JR2-wt). These results provide evidence that *VdPBP1* functions as a vesicle protein. However, it is important to note that the localization of the N-terminal of the protein has not yet been analyzed. 

### 3.3. The Deletion of VdPBP1 in V. dahliae Was Confirmed

To render the gene inactive, a putative ORF at the C-terminal end of the *VdPBP1* gene spanning 1.1 kb, known as *VdPBP1*#1 (Figure 3a), including the transmembrane helix signal, was deleted. The deletion process is illustrated in Figure 3a. The primary focus of this study was the examination of *VdPBP1*’s function, achieved by knocking out the 1.1 kb C-terminal region of the gene. The knockout strategy is depicted in Figure 3a.

The pKO2-*VdPBP1* plasmid was confirmed by PCR using the aforementioned primers and digestion with specified restriction enzymes before being introduced into the JR2-wt strain through the *A. tumefaciens* AGL-1 strain [18,22] in order to generate ∆*VdPBP1* strains (Table 3). The fungal transformants were then chosen on potato dextrose agar (PDA) that contained nourseothricin (50 µg mL^−1^) and cefotaxime (100 µg mL^−1^).

Before conducting functional analysis, the deletion strains of *VdPBP1* in *V. dahliae* were confirmed using Southern hybridization. The confirmation of *VdPBP1* deletion was performed using two different restriction enzymes, namely *Sac*I and *PshA*I. As a control, wild-type genomic DNA was utilized. The deletion strain exhibited bands of 3.8 kb and 4.2 kb, while the wild-type strain displayed bands of 9.6 kb and 3.4 kb (Figure 3b).

### 3.4. VdPBP1 Promote Microsclerotia Formation

*Verticillium* pathogens produce melanized microsclerotia, which serve as resting structures enabling their persistence in the soil for up to a decade, even without plant hosts. The influence of *VdPBP1* on the formation of microsclerotia was investigated using CDM plates containing 3% cellulose (Figure 4). Equal amounts of spores from JR2-wt, deletion, and complementation strains were inoculated onto a 3% cellulose (*w*/*v*) medium and cultivated at 25 °C for seven days. Microsclerotia present on the surface or within the agar were enumerated using binocular microscopy (Leica M165 FC) across three independent replicates.

Microsclerotia quantities on the surface and in the agar were quantified (Figure 4). The data depicted in Figure 4 indicate that the *VdPBP1* deletion strain exhibited an inability to form microsclerotia after seven days, while the wild-type strain displayed a substantial production of microsclerotia (Figure 4). This result remained consistent even when the culture time was extended to 14 days.

### 3.5. VdPBP1 Is Needed for Normal Hyphal Growth of V. dahliae on Agar Medium

The growth of the fungus on a nutrient medium serves as an indicator of its infectivity and pathogenicity when encountering the host. In this study, we conducted a comparison between the growth of wild-type and mutant mycelium on nutrient media to assess the impact of the *VdPBP1* gene. The obtained results are presented in Figure 5.

The findings in Figure 5 reveal that all mutants exhibited a 40% decrease in growth diameter compared to the wild type when cultured on the same medium. Furthermore, the aerial mycelium of the mutants displayed a slower growth rate compared to the wild type.

### 3.6. VdPBP1 Is Necessary for the Full Development of Disease Symptoms in Tomatoes

*V. dahliae* is a pathogenic fungus that causes wilting disease in over 200 plant species, including tomatoes [23,27]. In order to infect plants, *V. dahliae* enters through the roots and establishes colonization as the initial step of infection. To assess the impact of *VdPBP1* on virulence, an experiment was conducted using tomato plants as the host.

Ten-day-old tomato plants were selected to evaluate virulence in relation to the deletion of *VdPBP1*. Disease symptoms were compared between plants infected with the *VdPBP1* deletion strains and the wild-type strain over a period of 21 days post-infection (dpi) (Figure 6).

Tomato plants infected with the wild type exhibited a high percentage of disease index, reaching 81.67%, while uninfected plants (Mock) and plants infected with the *VdPBP1* mutant had percentages of disease indices of 1.6% and 20%, respectively (Figure 6a). However, severe disease symptoms were observed exclusively in tomato plants infected with the wild type. These symptoms included severe stunting, leaf chlorosis, and discoloration of the vascular system, as depicted in Figure 6.

However, it is noteworthy that the height of plants infected with the *VdPBP1* deletion strains was significantly higher than that of plants infected with the wild-type strains (Figure 6b). Nevertheless, the height of these plants was still significantly lower than that of mock plants. Uninfected plants did not exhibit black veins in the stems, whereas those infected with the wild-type and deletion strains did (Figure 6c).

To confirm the presence of the pathogenic fungus in the plants, re-isolation was performed from the hypocotyls [28]. The fungal pathogen was successfully re-isolated from the hypocotyls of plants infected with both the wild-type and *VdPBP1* deletion strains, while no re-isolation was possible from uninfected plants (Figure 6d).

### 3.7. VdPBP1 Regulates the Expression of Putative Target Genes for Microsclerotia Formation

In *Verticillium*, numerous genes have been identified to play a role in microsclerotia formation, including *SOM1*, *VTA2*, *VTA3*, *VTA1*, *SFl1*, *VHD1*, *VHD2*, *VHD3*, *VHD4*, and *VHD5*, some of which are crucial for pathogenicity, such as *SOM1*, *VTA2*, *VTA3*, and *PR1.* The expression of these genes in the *VdPBP1* mutant was assessed using RT-PCR. The results of the expression analysis revealed that only three genes, *VHD1*, *VTA1*, and *SFL1*, exhibited downregulation in the *VdPBP1* gene deletion strain (Figure 7). Specifically, the *VHD1* gene showed a reduction of over 90%, the *VTA1* gene was downregulated by 40%, and the *SFL1* gene displayed a 30% decrease in expression compared to the wild type. On the other hand, the expression levels of the other genes examined did not show significant changes when compared to the wild type. Based on these findings, it can be concluded that the *VdPBP* gene regulates the generation of microsclerotia by directly or indirectly controlling the expression of specific target genes, namely *VHD1*, *VTA1*, and *SFL1*.

### 3.8. Protein Regulatory Elements in the Promoter Region of the VdPBP1 Gene

In order to gain a better understanding of the role of the *VdPBP1* gene in regulating pathogenicity and microsclerotia formation, the promoter sequences of the gene were analyzed using the PROMO programs with the reference database of *S. cerevisiae* [29]. The analysis results of the 1.5 kb promoter region of the *VdPBP1* gene are depicted in Figure 8.

Figure 8 illustrates that the promoter region of the *VdPBP1* gene contains four CAAT boxes, three TATA boxes, and two GC boxes. Additionally, this promoter region contains four binding sites for the Rfx1 protein, two binding sites for the Mig1 protein, and one binding site for the Ste12 protein. These discernible binding sites potentially imply the presence of upstream proteins that partake in the modulation of *VdPBP1* gene expression within the pathway. Furthermore, these binding sites might provide insights into the cellular pathways in which the VdPBP1 protein is implicated.

## 4. Discussion

Quantifying microsclerotia quantities in the *VdPBP1* deletion strain and the wild-type strain provides valuable insights into the role of the *VdPBP1* gene in microsclerotia formation. This finding suggests that the *VdPBP1* gene plays a crucial role in the development or regulation of microsclerotia. Interestingly, the observed phenotype of the *VdPBP1* deletion strain aligns with the results obtained from the deletion of *SOM1*, as reported by Bui et al. [23]. This similarity implies a potential connection or interaction between Som1 and VdPbp1 in the same biological pathway. Since both genes exhibit a similar effect on microsclerotia formation when deleted, it suggests that they may function together or in a coordinated manner to regulate this process. The identification of shared phenotypic effects between the deletion of *SOM1* and *VdPBP1* provides a basis for further investigation into their functional relationship [23]. Future studies could explore the potential physical interaction or genetic regulatory interactions between these two genes. Understanding the molecular mechanisms underlying their collaboration could shed light on the specific pathways and processes involved in microsclerotia formation. Furthermore, the findings highlight the importance of considering multiple genes and their interactions in complex biological processes. Microsclerotia formation is a multifactorial trait influenced by various genetic components [1,2,23]. By uncovering shared phenotypes between different gene deletions, such as *SOM1* and *VdPBP1*, we can begin to unravel the intricate networks and pathways involved in regulating this process.

Furthermore, the observed phenotype of the *VdPBP1* deletion mutants aligns with the outcomes obtained from deleting the *VTA3* and *VTA2* genes within the same fungal species [3,23]. This consistency implies a potential functional relationship between *VdPBP1*, *VTA3*, and *VTA2* in the regulation of mycelial growth. The fact that the mutants lacking any of these genes exhibit similar growth defects strongly suggests that they are part of the same biological pathway or regulatory network. The identification of shared phenotypic effects between the deletion mutants of *VdPBP1*, *VTA3*, and *VTA2* provides a basis for further investigation into their functional interactions [3,23]. It would be valuable to explore the potential genetic or molecular connections between these genes to understand how they collectively contribute to mycelial growth regulation. Furthermore, these findings highlight the complexity of the molecular mechanisms underlying mycelial growth. Multiple genes and pathways are likely involved in coordinating this process, and the deletion of key genes such as *VdPBP1*, *VTA3*, and *VTA2* disrupts the regulatory network, resulting in the observed growth defects. 

The results obtained from infecting tomato plants with the wild-type and *VdPBP1* deletion strains reveal interesting findings regarding the role of the *VdPBP1* gene in the pathogenicity of *V. dahliae*. Comparisons with previous experiments involving the deletion of the *SOM1* and *VTA3* genes in tomato plants highlight the differences in outcomes [23]. In those cases, the deletions did not induce disease symptoms in infected plants, and the fungal pathogen could not be re-isolated from the stems [23]. However, in the present study, although the deletion of the *VdPBP1* gene did not result in typical disease markers, black marks were observed in the stem segments, and the fungus could be easily isolated from them. This suggests that the *VdPBP1* gene is involved in host pathogenicity rather than mere infection. These findings imply that *VdPBP1* plays a crucial role in the virulence of *V. dahliae* in tomato plants. Further investigations are needed to unravel the specific mechanisms underlying the pathogenicity conferred by *VdPBP1* and to explore its potential interactions with other genes involved in host–pathogen interactions. Understanding the function of *VdPBP1* will contribute to our knowledge of the molecular basis of fungal pathogenesis and may facilitate the development of effective strategies to control *Verticillium* wilt in tomato crops.

The results presented in this study provide compelling evidence supporting the role of the *VdPBP1* gene in regulating the generation of microsclerotia in *V. dahliae*. The data indicate that the deletion of the *VdPBP1* gene leads to an inability to form microsclerotia, as demonstrated by the lack of microsclerotia observed after seven days. In contrast, the wild-type strain showed substantial production of microsclerotia. This clearly indicates that the *VdPBP1* gene is essential for the formation of microsclerotia in *V. dahliae*. Moreover, the similarities observed in the expression analysis of the *SOM1* and *VTA3* gene deletions further support the notion that these genes, along with *VdPBP1*, operate in the same pathway during microsclerotia formation [23]. The findings suggest that Som1, Vta3, and VdPbp1 may interact and regulate the expression of specific target genes, such as *VHD1*, *VTA1*, and *SFL1*, which are crucial for microsclerotia development. These results contribute to our understanding of the molecular mechanisms underlying microsclerotia formation in *V. dahliae* and highlight the potential interplay between different genes in this process. The identification of shared phenotypic effects and the parallel expression patterns among these gene deletions suggest their involvement in a common pathway or regulatory network. Further investigations are necessary to unravel the precise mechanisms by which Som1, Vta3, and VdPbp1 interact and regulate the expression of target genes during microsclerotia formation. It would be valuable to explore the genetic and molecular connections between these genes, elucidating the signaling pathways and transcriptional regulatory networks involved.

The analysis of the promoter region of the *VdPBP1* gene (Figure 7) provides valuable information regarding the regulatory elements and protein interactions involved in the expression of this gene. The presence of multiple CAAT boxes, TATA boxes, and GC boxes suggests the involvement of general transcription factors in initiating transcription from this promoter region. Of particular interest are the binding sites for specific proteins, namely Rfx1, Mig1, and Ste12. The identification of four binding sites for Rfx1, which is known to be a homolog of Vta3, suggests a direct interaction between Vta3 and the promoter of the *VdPBP1* gene [23]. This implies that the Vta3 protein likely regulates the expression of *VdPBP1*, which is consistent with the role of Vta3 in microsclerotia formation. Similarly, the presence of Mig1-binding sites suggests a potential involvement of the Vta6 protein, a homolog of Mig1, in the regulation of the *VdPBP1* gene [27,30]. This indicates that Vta6 might also influence microsclerotia formation and pathogenicity by binding to the promoter region of *VdPBP1*. The connection between the Ste12 protein and the Flo8 protein in *S. cerevisiae* provides insights into the potential role of Som1, a homolog of Flo8, in regulating the expression of the *VdPBP1* gene [23,31]. As Som1 is known to be involved in the initiation of infection, disease development, and the regulation of microsclerotia and conidia production, it is plausible that Som1 directly or indirectly controls the expression of *VdPBP1*, thereby influencing pathogenesis and microsclerotia formation.

Overall, this study highlights the intricate nature of gene regulation in *V. dahliae* and the significance of specific protein interactions and regulatory elements in governing fungal growth, pathogenesis, and microsclerotia formation. The knowledge gained from these findings can contribute to the development of targeted strategies for controlling *Verticillium* wilt and other fungal diseases affecting important crops.

## 5. Conclusions

The study focused on the characterization of the *VdPBP1* gene in *V. dahliae* and its role in fungal growth and pathogenesis. The gene was found to encode proteins containing both an LsmAD domain and a PAB-binding domain. Bioinformatic analysis revealed the presence of a 9aaTAD domain, DNA-binding sites, a potential nuclear localization signal (NLS), and a transmembrane helix signal in the *VdPBP1* protein sequence. VdPBP1 is located within vesicles. Deletion of *VdPBP1* in *V. dahliae* confirmed the gene’s structure, resulted in the loss of microsclerotia formation, and reduced hyphal growth, percentage of disease index, and disease symptoms in tomato plants. The expression analysis showed that *VdPBP1* regulates the expression of specific target genes involved in microsclerotia formation. The *VdPBP1* promoter contains binding sites for transcription factors, suggesting a potential interaction between previously studied transcription factors and the regulation of *VdPBP1* gene expression. Overall, these findings highlight the importance of *VdPBP1* in fungal development, pathogenicity, and the regulation of key genes associated with microsclerotia formation in *V. dahliae*. Further investigations into the molecular mechanisms and interactions of *VdPBP1* will contribute to a deeper understanding of fungal pathogenesis and potentially lead to the development of novel control strategies for plant diseases caused by *Verticillium* pathogens.

## Figures and Tables

**Figure 1 microorganisms-12-00265-f001:**
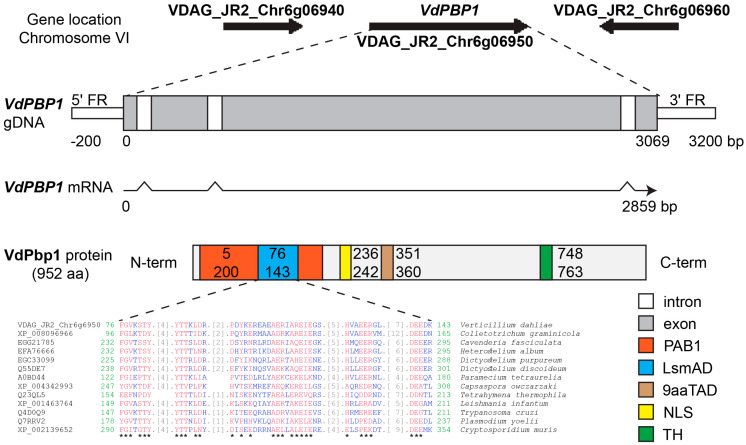
Gene locus and structure of *VdPBP1*. Genomic DNA was compared with cDNA using Multalin. Introns and exons were marked. The nuclear localization signals (NLS) predicted by the SeqNLS software are displayed. The protein domains were predicted by three different programs: Pfam, NCBI blast, and 9aaTAD. Sequence alignments of the LsmAD domain of VdPbp1 and related proteins from other organisms including *Colletotrichum graminicola* (XP_008096966), *Cavenderia fasciculata* (EGG21785), *Heterostelium album* (EFA76666), *Dictyostelium purpureum* (EGC33099), *Dictyostelium discoideum* (Q55DE7), *Paramecium tetraurelia* (A0BD44), *Capsaspora owczarzaki* (XP_004342993), *Tetrahymena thermophila* (Q23QL5), *Leishmania infantum* (XP_001463764), *Trypanosoma cruzi* (Q4D0Q9), *Plasmodium yoelii* (Q7RRV2), and *Cryptosporidium muris* (XP_002139652) are displayed. Asterisks and red: identical residues, blue: low (50%) consensus values; green: the start and end site of alignments sequence; FR: flanking region; N-term: N-terminus; C-term: C-terminus; PAB1: PAB1-binding domain; LsmAD: LsmAD domain; 9aaTAD: nine-amino-acid transactivation domain; NLS: nucleus localization signal; TH: transmembrane helix.

**Figure 2 microorganisms-12-00265-f002:**
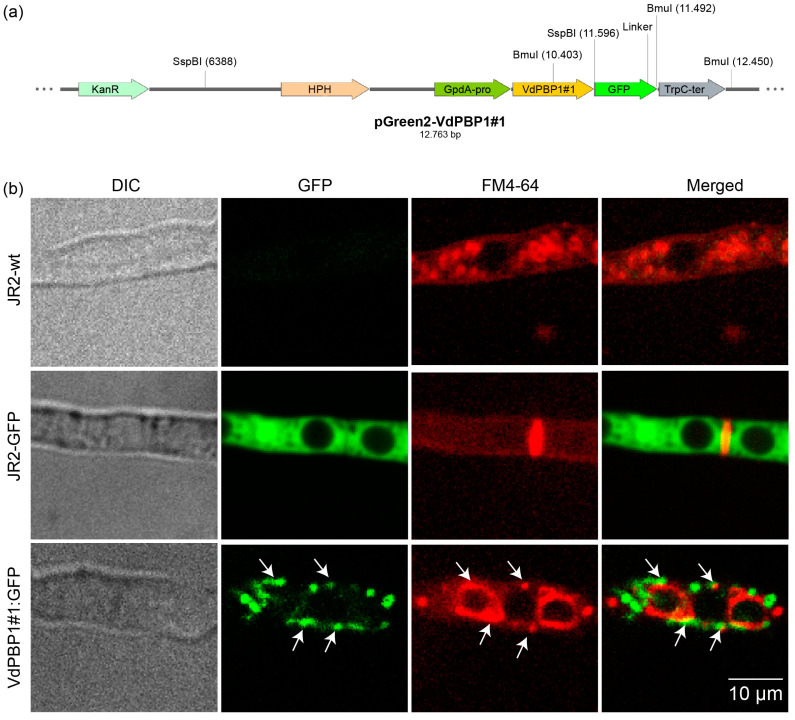
VdPbp1 is a vesicle protein. The *VdPBP1#1* fragment, driven by the GpdA promoter, was fused to *GFP*, separated by the GGSGG linker, and introduced into the JR2-wt strain through ectopic transformation. Transformants were selected on selective medium containing hygromycin (50 µg ml^−1^) and then examined using fluorescence microscopy. Positive clones displaying a strong GFP signal were cultured in liquid SXM for 24 h. The localization of proteins was observed through green fluorescence, while fungal vesicles were visualized with FM4-64 (N-(3-Triethylammoniumpropyl)-4-(6-(4-(Diethylamino) Phenyl) Hexatrienyl) Pyridinium Dibromide) staining and indicated by red color. The strain names and filters used are indicated. Scale bars are illustrated. Arrowheads indicate vesicles. (**a**) The pGreen2-VdPBP1#1 construct (**b**) The localization of VdPBP1 proteins.

**Figure 3 microorganisms-12-00265-f003:**
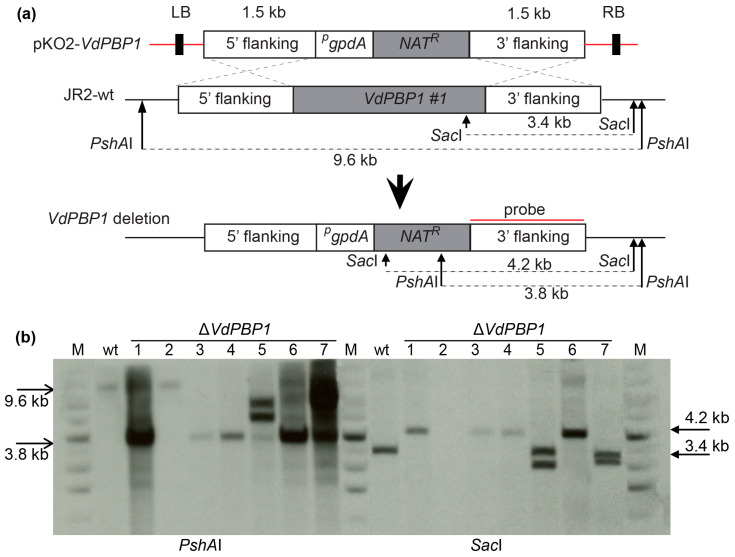
Verification of *VdPBP1* deletion strains of *V. dahliae*. (**a**) Strategy for deletion of the *VdPBP1* gene via homologous recombination between the deletion construct and *V. dahliae* JR2 wild-type (JR-wt) *VdPBP1* locus. The regions 1.5 kb upstream and 1.5 kb downstream of the ORF *VdPBP1#1* are denoted as 5′ flanking and 3′ flanking regions, respectively; p: promoter; R: resistance; t: terminator. (**b**) Confirmation of deletion strains by Southern hybridization with two different restriction enzymes *Sac*I and *PshA*I. The probe is the 3′ flanking region. Genomic JR2-wt DNA was used as a control.

**Figure 4 microorganisms-12-00265-f004:**
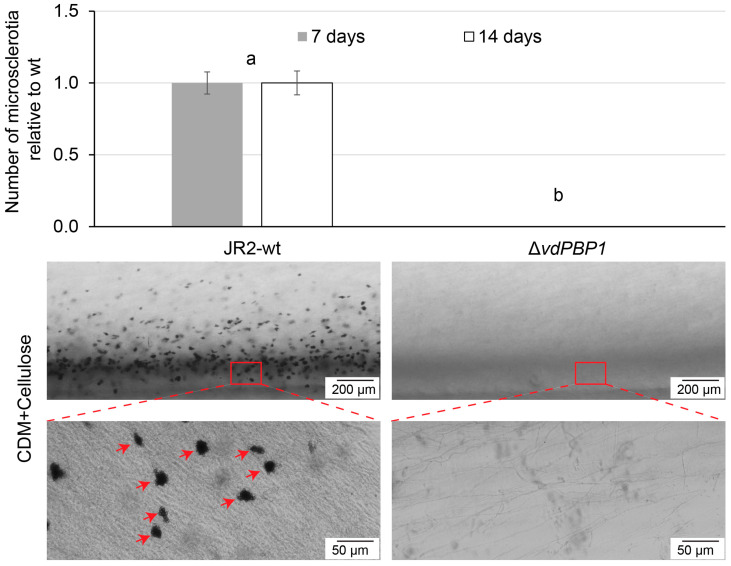
*VdPBP1* supports microsclerotia formation. The same number of spores of JR2-wt *VdPBP1* deletion strains were dropped on CDM plates containing 3% cellulose and were incubated in the dark. The arrowhead indicates a microsclerotium. The number of microsclerotia in the same area was counted. Experiments were performed in triplicate. The mean values and standard deviation are shown. Gray columns and white columns represent data 7 days and 14 days after culture. Letters a, b indicate groups of significant differences as calculated by Tukey–Kramer multiple comparison procedures, α = 0.01. Scale bars are displayed.

**Figure 5 microorganisms-12-00265-f005:**
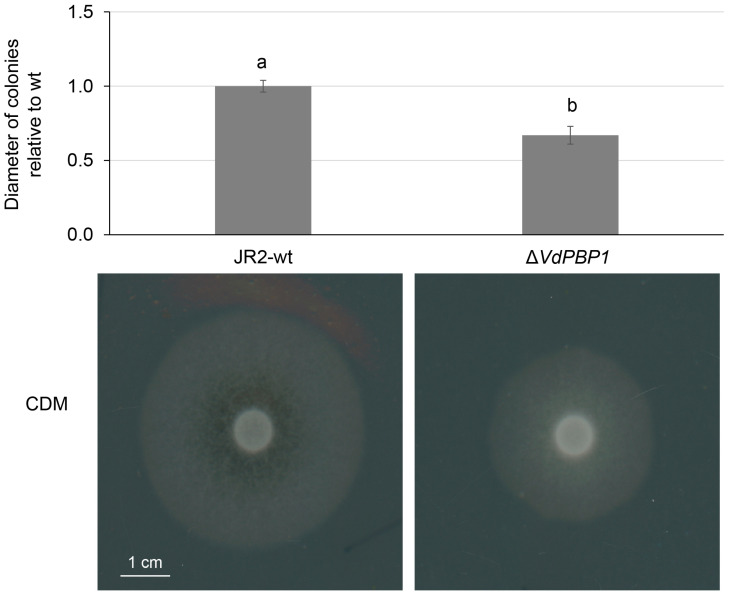
*VdPBP1* is required for hyphal development of *V. dahliae*. The same number of spores of JR2-wt *VdPBP1* deletion strains was dropped on CDM plates and grown in the dark at 25 °C for ten days. The development of aerial hyphae was observed on the surface of plates. Average values and standard deviation are shown. The letters a and b indicate the groups that are significantly different, calculated by Tukey–Kramer multiple comparison procedures, α = 0.01. Scale bars are displayed.

**Figure 6 microorganisms-12-00265-f006:**
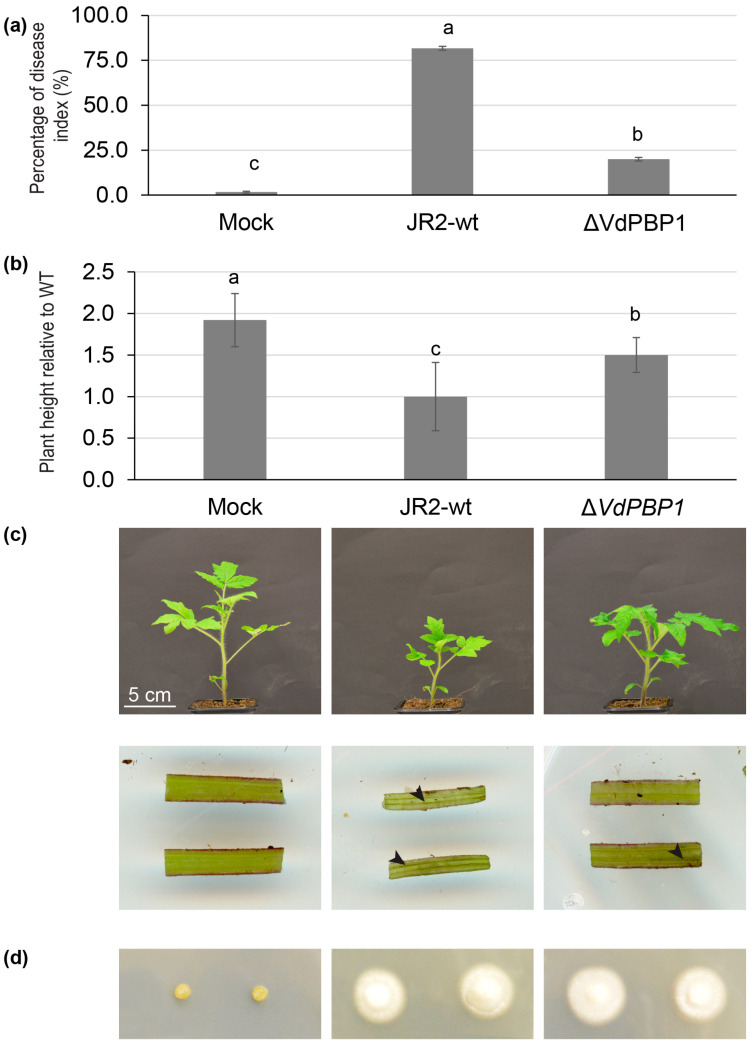
*VdPBP1* is required for introduction of disease symptoms in tomatoes. Ten-day-old tomatoes were infected with the same number of spores of indicated strains or remained uninfected (mock). The plants were incubated in the climate chamber under 16 h light/8 h dark at 22–25 °C. The disease symptoms were assessed at 21 days post-infection (dpi). Infection experiments were performed with 12 single plants for each fungal strain. (**a**) The percentage of disease index. (**b**) The plant height was measured. The mean values and standard deviations are indicated. The letters a, b and c show groups that are significantly different as calculated by Tukey–Kramer multiple comparisons procedure, α = 0.01. Representative plants are shown. Scale bars are indicated. (**c**) Black threads in infected stems were observed. The arrowhead indicates black veins. (**d**) The process of fungal re-isolation from hypocotyls was assessed. After surface sterilization, the hypocotyls were positioned on plates of potato dextrose broth medium (PDM) containing cefotaxime and chloramphenicol. Subsequently, they were incubated for a duration of 7 days.

**Figure 7 microorganisms-12-00265-f007:**
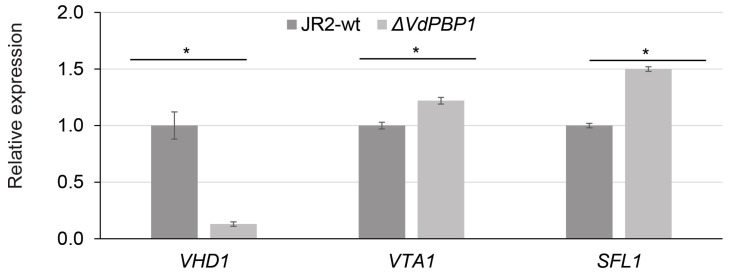
*VdPBP1* regulates the expression of putative target genes for microsclerotia formation. RNAs of *V. dahliae* JR2-wt *VdPBP1* deletion strains were harvested from 3-day-old mycelia grown in 50 mL liquid SXM. The same amount of RNA was used for cDNA synthesis. The expression of putative target genes was normalized to a housekeeping gene (GAPDH encoding glyceraldehyde-3-phosphate dehydrogenase) and the wild-type strain. The mean values and standard deviations of 4 repetitions are presented. * represents significant differences compared to wild type as calculated by Tukey–Kramer comparison procedures, α = 0.01.

**Figure 8 microorganisms-12-00265-f008:**
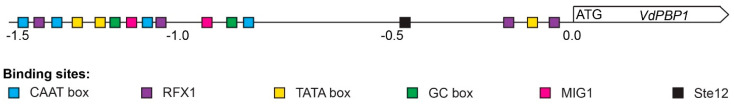
Schematic outline of *VdPBP1* promoter regions with unique transcription factor-binding sites. Binding sites of transcription factors on the promoter region of *VdPBP1* were identified with the PROMO program. Transcription start site of genes (TSS) was predicted based on the cDNA sequence and protein-coding sequence of VdJR2 in the Ensembl Fungi database. A 1500 bp part of the *VdPBP1* promoter regions was analyzed and the TATA box within the core promoter, the CCAAT box, and the GC box of promoter-proximal elements as well as the Rfx1, Mig1, and the Ste12 sites are highlighted.

**Table 1 microorganisms-12-00265-t001:** Fungal strains used in this study.

*V. dahliae* Strain Name	Description	Reference
JR2-wt	JR2-wt	[19]
∆VdPBP1	*∆VdPBP1:NAT1* ^R^	This study

^R^: resistance, NAT1: nourseothricin.

**Table 2 microorganisms-12-00265-t002:** Plasmids used in this study.

Plasmid Name	Description	Reference
PKO2	^p^ *trpC*:*NAT1* ^R^	[23]
pKO2-*VdPBP1*	^p^ *VdPBP1*:^p^ *trpC*:*NAT1* ^R^: *VdPBP1* ^t^	This study
pGreen2-*VdPBP1*#1	^p^ GpdA:*VdPBP1*#1::*GFP*:trpC ^t^	This study

^p^: promoter, ^t^: terminator, ^R^: resistance, NAT1: nourseothricin.

**Table 3 microorganisms-12-00265-t003:** Primers used in this study.

Primer Names	5′-3′ Direction	Reference
*gdpA-NAT-F*	*CAACTGATATTGAAGGAGCATTT*	This study
*gdpA-NAT-R*	*TCAGGGGCAGGGCATGC*	This study
*VdPBP1*-P1 (*Pac*I)	TACGAATTCTTAATTAACTCGACCCGTGAAAACAAGT	This study
*VdPBP1*-P2 (*Spe*I)	GTACCACTAGTGAGCTCACGCCACGTTGGTATCGTAT	This study
*VdPBP1*-P3 (*Xba*I)	TCGAGAGGCCTTCTAGAGGTGGTTCTAGGCTGCATGT	This study
*VdPBP1*-P4 (*Sbf*I)	GCTTGCATGCCTGCAGGAATCACCTCGTCGCAGTCAC	This study
*OE-VdPBP1-F*	* ACCCTGACATCACCCTCGAG * *ATGCGCCTGACATACAAAGAA*	This study
*OE-VdPBP1-R*	CCTCGCCCTTGCTCACCAT**ACCACCGCTACCACC**CTTGGCCTCGGCAGGCC	This study

Words with underlines: nucleotide overhang; bold words: the linker.

## Data Availability

Data are contained within the article.

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
