# Peer review of "Verticillium dahliae VdPBP1 Transcription Factor Is Required for Hyphal Growth, Virulence, and Microsclerotia Formation"

_microorganisms, 2024, doi:10.3390/microorganisms12020265_

Round 1

Reviewer 1 Report (Previous Reviewer 2)

Comments and Suggestions for Authors

Most of my concerns have been addressed properly. 

Author Response

Thank you for your valuable comments.

Reviewer 2 Report (Previous Reviewer 1)

Comments and Suggestions for Authors

It can not be concluded that VdPbp1 is a transmembrane pore protein through Fig. 2. Furthermore, the staining with FM4-64 is not good, and is very different among those three strains. The views of DIC are also not good. Why not use a full length VdPbp1 to do the subcellular localization assay?

For the column chart in Fig. 4, the gray and white columns should also be explained in the chart, not only in the figure legend. Why not include the date from 14 days’ culture together in the column chart since the authors already did that? In lower panel of Fig. 4, the scale bars are incorrect in the magnified pictures.

In Fig. 6a, why a mock inoculation had a disease index?

In Fig. 6c, I think the reduced plant height and black veins are the symptoms in the deletion mutant, so it is incorrect in Line 373 and 374 and conclusion.

In 3.3, detailed methods should be moved to 2.4.

The title is not suitable. Hyphal growth and virulence are not blocked in VdPBP1 deletion mutant.

The title of 3.6 is incorrect.

Comments on the Quality of English Language

Need some editing. 

Author Response

Thank you for your valuable comments. We have addressed all your comments in the response letter and revised the manuscript accordingly. Please see the attachment.

Round 2

Reviewer 2 Report (Previous Reviewer 1)

Comments and Suggestions for Authors

There are improvements in the revised manuscript. However, I still concern in some aspects.

1.        I am still not convinced about the subcellular localization of VdPBP1#1. My question is that why the authors considered its membrane pore localization? As far as I know, FM4-64 stains the cell membrane, septa, and vesicles. Could the authors provide some references or protocols to show in which situation FM4-64 could stain the cell pores? Or add a marker protein in the figure that has been proved to localize in the cell membrane pores. Why to conclude that the arrowheads indicate membrane pores?

2.        In lower panel of Fig. 4, the scale bars are still incorrect in the magnified pictures. They are not only 4 times magnified. Please check carefully.

3.        Figure 6a, disease index should be a number, not a percentage.

4.        I understand sometimes plants do not grow well, but the disease index is incorrect according to your response (one in Grade One, 11 in Grade 0). The disease index of Mock should be much lower, about 2. Please check and calculate again.

5.        Remove the frame of bar chart in Figure 5.

6.        Adjust Figure 4 and Figure 7 in same format. Eg, the position of legends (7 days, 14 days vs JR2, ΔVdPBP1).

7.        A “.” is missed in Line 496.

8.        Some references are not formatted well.

9.        Check whether the pictures in Figure 5 have been stretched, since the colonies are not perfectly round.

10.    Check the format of all the bar charts. They are not in same format. Eg, there are frame out the columns or not.

Comments on the Quality of English Language

Minor editing of English language required

Author Response

Thank you for your valuable comments. Please see the attachment.

This manuscript is a resubmission of an earlier submission. The following is a list of the peer review reports and author responses from that submission.

Round 1

Reviewer 1 Report

Comments and Suggestions for Authors

Comments are listed below.

1. L202, should be sequence contains.

2. In Fig. 2b, it is not clear that the VdPBP1 localized in the transmembrane pore protein, where may need more delicate observation or marker to conclude. Besides, why did authors indicate that N-terminal has not been analyzed in L241. Since this assay used the full length protein, the GFP signal showed the localization of the full protein, not only the C-terminal of the protein. And, all strains should show the same part of the hypha with similiar morphology in Fig. 2. 

3. According to Fig. 3, full length VdPBP1 gene is deleted. But why the authors indicated that only 1.1 kb fragment was deleted?

4. L302, should be Figure 4.

5. Characters in Bars in Figure 4 are too small to be seen.

6. L326, needed is not accurate, because deletion of this gene also enable the fungi to grow. Should be important or needed for normal growth or sth.

7. Figure 6, there already are symptoms in the deletion mutant, since plant height are much lower than the mock. Moreover, a disease index or biomass analysis is needed. 

8. Nealy all promoters are bound by transcription factors, so it will never suggest a complex regulatory network, in L559.

Comments on the Quality of English Language

Some words are not accurately used, L333 should be slower growth or lower growth rate, for example. Words should be spelled consistently in the whole text, such as localisation.

Reviewer 2 Report

Comments and Suggestions for Authors

In this resubmission, the authors have addressed most of my concerns properly.

However, I am not satisfied with the change in Fig3a. I suggest that the authors first depict the entire VdPBP1, then highlight where the 1kb deletion is located in VdPBP1. In the current Fig3a, it is still hard to tell whether you deleted the whole gene or just part of it.

Additionally, I am also concerned about the new fluorescence microscopy:

1. Replace the "RFP" label with FM4-64; FM4-64 is not RFP. Showing the filter is not as meaningful as labeling the exact stain

2. Why is the distribution pattern for FM4-64 so different in JR2-wt, JR2-GFP, and VdPBP#1:GFP?

3. Please select a similar structure for imaging; the structure presented in JR2-wt and VdPBP1#1:GFP looks distinct.

4. Different from what is written in Line 237, I can barely see the co-localization between FM4-64 and the GFP signal. Therefore, I would like to suggest the authors re-evaluate their conclusion from the microscopy.

Comments on the Quality of English Language

Minor editing of English language required